# Synthesis of Calcium Aluminates from Non-Saline Aluminum Dross

**DOI:** 10.3390/ma12111837

**Published:** 2019-06-06

**Authors:** Félix Antonio López, María Isabel Martín, Francisco José Alguacil, Mario Sergio Ramírez, José Ramón González

**Affiliations:** 1Centro Nacional de Investigaciones Metalúrgicas (CENIM-CSIC), Avda. Gregorio del Amo, 8, 28040 Madrid, Spain; mariaimh@hotmail.com (M.I.M.); fjalgua@cenim.csic.es (F.J.A.); 2ARZYZ, S.A. of C.V Avda, Don Mario Sergio Ramírez Morquecho 794, Pesquería River, 66632 Cd Apodaca, N.L., Mexico; marios.ramirez@arzyz.com (M.S.R.); jose.gonzalez@arzyz.com (J.R.G.)

**Keywords:** aluminum, non-saline dross, tricalcium aluminate, calcium aluminates, reactive milling, sintering

## Abstract

The present work examines the synthesis of tricalcium aluminate (for use as a synthetic slag) from the non-saline dross produced in the manufacture of metallic aluminum in holding furnaces. Three types of input drosses were used with Al_2_O_3_ contents ranging from 58 to 82 wt %. Calcium aluminates were formed via the mechanical activation (reactive milling) of different mixtures of dross and calcium carbonate, sintering at 1300 °C. The variables affecting the process, especially the milling time and the Al_2_O_3_/CaO molar ratio, were studied. The final products were examined via X-Ray diffraction (XRD), scanning electron microscopy (SEM), transmission electron microscopy (TEM) and Raman spectroscopy. The reactive milling time used was 5 h in a ball mill, for a ball/dross mass ratio of 6.5. For a molar relationship of 1:3 (Al_2_O_3_/CaO), sintered products with calcium aluminate contents of over 90% were obtained, in which tricalcium aluminate (C_3_A) was the majority compound (87%), followed by C_12_A_7_ (5%).

## 1. Introduction

The calcium aluminates are described in the CaO–Al_2_O_3_ binary phase diagram [1,2]. Within this system, five binary compounds can be distinguished that generically go by the name of calcium aluminates: CaAl_2_O_4_ (CA), CaAl_4_O_7_ (CA_2_), Ca_12_Al_l4_O_33_ (C_12_A_7_), Ca_3_AlO_6_ (C_3_A), and CaAl_12_O_19_ (CA_6_), where C = CaO and A = Al_2_O_3_.

One of the mineral phases that constitute Portland cement is tricalcium aluminate (Ca_3_Al_2_O_6_), which plays an important role in the cement setting process, especially in the first stages of the hydration process [3]. One of the more common processes for the synthetizing of C_3_A is by way of a solid–solid state reaction between CaO and Al_2_O_3_, or by the thermal decomposition of CaCO_3_ and Al(OH)_3_ [4,5]. There is no unanimity in the literature regarding the mechanisms of its production. Many reactions are possible among the calcium and aluminum oxides that lead to different calcium aluminate phases, among them C_3_A, C_12_A_7_, CA, CA_2_, and CA_6_, which can react with one another or with the calcium and aluminum oxides, forming new aluminates. A few of the reactions suggested by Singh et al. [6] for making C_3_A are

CA (s) + 2CaO (s) → C_3_A (s),(1)

CA_2_ (s) + 5CaO (s) → 2C_3_A (s),(2)

C_12_A_7_ (s) + 9CaO (s) → 7C_3_A (s),(3)

C_12_A_7_ (s) + CA (s) + 11CaO (s) → 8C_3_A (s).(4)

Kuzmenko et al. [7] also suggest that tricalcium aluminate is produced through Reaction 1 by the diffusion of Ca^2+^ in an alumina network. However, there is controversy regarding the production of C_3_A from the intermediate CA and C_12_A_7_ phases. Some authors [8] identified phases such as C_2_A, C_12_A_7_, CA, CA_2_, and CA_6_ during C_3_A synthesis, although they concluded that only the CA_2_ and C_2_A phases were intermediate products or precursors of the reaction that produces tricalcium aluminate. Other authors suggest that the CA and C_12_A_7_ phases are intermediate phases in tricalcium aluminate synthesis [9]. Lastly, Ghoroi and Suresh [4] suggest a fast conversion of the alumina and the diffusion of Ca^2+^ to generate the intermediate C_12_A_7_ phase that slowly evolves afterward into the final C_3_A product, again, by the diffusion of Ca^2+^ in the alumina network.

Other C_3_A synthesis processes described in the literature are based on the use of the method Gaki et al. [10] or a modification of this method to reduce the process stages and the sintering temperature [11]. In this modified synthesis it is possible to synthesize C_3_A between 1300 and 1350 °C with reaction times of 1 to 4 h. Zivica et al. [12] synthesized C_3_A by several successive firings of a pressed molar mixture of CaCO_3_ and Al_2_O_3_ at a temperature of 1400 °C with a cooking time of 5 h. Salimi and Vaughan [13] synthesized C_3_A by priming slaked lime with a sodium aluminate solution at 368 K in a continuous stirred-tank reactor. Finally, other alternatives have been proposed for the synthesis at low temperature based on sol–gel and combustion techniques [14,15]. Pure compounds are used to synthesize tricalcium aluminate in all of the synthesis processes described.

However, there are some patents that describe how different types of aluminates may be obtained from aluminum industrial waste. Beelen and Willen Van [16] developed a process of obtaining calcium aluminates through two consecutive steps: First, the treatment of dross to recover aluminum, and second, treatment of the resulting dross with CaOH_2_, followed by calcination. Kemey et al. [17] describe obtaining calcium aluminate from aluminum dross, irrespective of their composition and fluxes, based on SiO_2_ and/or of CaO by fusion at 1470 °C. Pickens and Morris [18] describe the preparation of calcium aluminates from aluminum dross with calcium oxide and/or precursors of CaO, using a bonding agent at a temperature between 2000 and 2300 °C. Finally, the Spanish patent IS 2343052 B2 [19] describes obtaining calcium aluminates from the residue obtained after the treatment of salt dross from the production of secondary aluminum, using calcium oxide and/or precursors of CaO.

There are also some laboratory-level studies that synthesize calcium aluminates using as starters different residues containing alumina. Ewais et al. [20] use different mixtures of sludge and aluminum dross to make calcium aluminates in a temperature range between 1250 and 1550 °C. Li et al. [21] have used aluminum dross as raw material to prepare refractory materials of high alumina content and have obtained refractory materials of one principal crystalline phase (MgAl_2_O_4_) and small amounts of CaAl_4_O_7_ at a temperature of 1530 °C. López-Delgado et al. [22] describe the synthesis of calcium aluminates C_12_A_7_, CA_2_, and CA from the hazardous waste of the tertiary aluminum industry, using a precursor obtained by hydrothermal method and subsequent heat treatment in differential thermal analysis and thermal gravimetric analysis (DTA/TGA). Fernández et al. [23] describe the synthesis of different aluminates (CA_2_, CA_6_ and CA) with alumina by synthesizing a mix of alumina, calcium carbonate, and charcoal at 1400 °C, employing a solar concentrator.

Taking into account that the CaO–Al_2_O_3_ binary system shows that C_3_A fuses incongruently to 1542 °C [24], the calcium aluminates have applications in heat-resistant cement since they are stable at high temperatures. They similarly have applications in steelmaking, in which the provision of a synthetic, calcium aluminate-based dross favors the desulfuring of the steel and the production of steel free of inclusions (especially of Al_2_O_3_) [25]. The presence of a molten calcium aluminate slag on the steel (i.e., synthetic slags) also facilitates secondary metallurgical work via its positive influence on the fluidity of the steel, its protection against re-oxidation, and via the prevention of temperature loss [26]. Most of the calcium aluminate used in the steel sector is sintered from mixtures of bauxite and lime. The Al_2_O_3_-rich dross produced during the melting of aluminum can be used as an alternative to bauxite [27].

In this work, a solid-state synthesis route has been followed at a temperature of 1300 °C but, prior to sintering, the dross and limestone mixtures underwent a reactive grinding process (mechanochemical process). This treatment increases the reactivity of limestone and of dross as a result of the changes produced in the material structure by grinding (disorder, relaxation, and mobility) because of the applied mechanical energy. This occasions bond rupturing, which generates high-energy zones, originating fractures and new surfaces, all of which facilitates solid-state reactions [28,29,30]. Additionally, some projects describe calcium aluminate synthesis by way of mechanical activation [31,32]. As a result of mechanical activation, solid-state reactions are faster and occur at lower temperatures.

This work describes the synthesis of calcium aluminate, through solid-state reactions, from mixing limestone from different non-saline drosses from the manufacture of metallic aluminum in holding furnaces. The end products were examined via XRD, SEM, TEM, and Raman spectroscopy.

## 2. Materials and Methods

### 2.1. Dross

Three samples of dross have been used that correspond to different periods of storage of the dross produced in the metal-aluminum fusion plant from which they come.

Sample Al-1: Dross aged 3 to 7 years.Sample Al-2: Dross with an age of 7 to 10 years, stored outdoors.Sample Al-3: Recent dross, generated in the last two years.

The drosses were quartered and dried in a stove (80 °C/24 h) and ground up for 15 min in a cylinder mill until materials with a particle size of less than 40 µm were obtained.

### 2.2. Characterization 

The chemical composition of dross and of sintered products was determined by inductive coupling plasma spectroscopy, using an inductively coupled plasma atomic emission spectroscopy (ICP–OES) Varian 725-ES, Agilent Technologies, Santa Clara, CA, USA). Previously, the samples were attacked with Metaborate Lithium (Merck KGaA, Damstadt, Germany) at 1050 °C and acidified with concentrated nitric acid (HNO_3_). At the same time, losses were determined by calcination according to ISO 1171:2010 (815 °C/1 h).

The mineral composition of the drosses and of the sintered products was obtained by X-ray diffraction making use of a Siemens D5000 diffractometer (Siemens, Munich, Germany) equipped with a Cu anode (Cu Kα radiation) and a LiF monochromator to eliminate the Kβ radiation of samples containing iron. The generator’s voltage and current were 40 kW and 30 mA, respectively. The measurement was carried out continuously at 0.03° and at a rate of 3 s for each step. The diffractograms were interpreted with the help of the ICDD (International Centre for Diffraction Data) Powder Diffraction File (PDF-2) reference database and the Bruker AXS DIFFRACplus EVA software (vs. 4.3, Bruker GmbH, Karlsruhe, Germany). The X-ray diffraction diagrams were used to perform a quantitative study of the crystalline phases present in the drosses and in the sintered products through the Rietveld method. XRD data refinement was accomplished using the Bruker AXS Rietveld Topas analysis program.

The microstructural analysis of the samples was performed by field emission scanning electron microscopy (FESEM) in a HITACHI S-4800 (Tokyo, Japan), using a voltage of 15 kW. The microscopy samples were stuffed in a polymeric resin and polished with 600-, 1200- and 2000-grit sandpaper (adding carnauba to these to protect the sample). Afterward, the samples were polished with 3 and 1 µm diamond paste and were metallized with carbon in a JEOL JEE 4B (Tokyo, Japan).

A sample consisting mainly of tricalcium aluminate, obtained in the best operating conditions, was also studied by both Transmission Electron Microscopy (TEM) using a JEOL JEM 2100 HT (Tokyo, Japan), as well as by Raman Microscopy using a Jobin-Yvon LabRAM HR800 Horiba confocal microscope (Horiba, Kyoto, Japan). The samples were excited by a 633 nm He–Ne laser on an Olympus BX41 confocal microscope (Tokyo, Japan) with a 10× objective.

### 2.3. Aluminate Preparation

The three dross samples were mixed with CaCO_3_ in different molar proportions (Al_2_O_3_/CaO 1:1, 1:2, and 1:3). A PanReac CaCO_3_ reagent (Panreac Química, Darmstadt, Germany) of PA (precipitated for analysis) quality with a minimum purity of 99.0% was used. These dross mixtures were subjected to reactive grinding for 5 h in a Fritsch Pulverisette 6 mill (Fritsch, Idar-Oberstein, Germany) at 450 rpm, with 5 stainless steel balls with a ball weight/mixture ratio of 6.54. At the completion of the grinding period, cylindrical mini briquettes (13.5 mm (diameter) × 5.5 mm (height)) were prepared without addition of binding agents by compacting in a Specac 15 T Atlas manual hydraulic press (Specac Ltd., Orpington, UK) at 1034 MPa pressure. To achieve the complete breakdown of the calcium carbonate, the mini briquettes first underwent isothermic treatment in an EVA electric oven (Linn High Therm, Eschenfelden, Germany) at 750 °C for 1 h, then further treatment at 1300 °C for 1 h.

## 3. Results and Discussion

### 3.1. Chemical Composition

The chemical composition of the drosses are shown in Table 1. Drosses Al-1 and Al-3 present similar chemical compositions, whereas dross Al-2 has a lower Al content and a higher percentage of Zn. The losses by calcination, which include moisture, interstitially absorbed water, mineral phase water of crystallization, and mineral phase decomposition, each present very different values.

Figure 1 shows the XRD diffraction pattern of the drosses studied. It is observed that the oldest drosses (Al-1 and Al-2) have a greater amorphous character than the recent dross (Al-3), which clearly presents a higher degree of crystallinity.

Table 2 shows the quantitative mineral composition of each sample, calculated by the Rietveld method. Samples Al-1 and Al-3 have a similar phase composition. In dross Al-2, boehmite and gibbsite are present, which do not appear in the other two drosses, while there is no presence of norstrandite, enstatite, or magnesite phases, nor of Mg spinel. Sample Al-2 is more hydrated than the other two, possibly due to having been stored outdoors for years.

The total content in Al and Ca hydrates varies in order: Al-2 (62%) > Al-1 (9.13%) > Al-3 (5.95%), which is the same order in which losses vary by calcination.

### 3.2. Chemical Composition and Microstructural Characterization of Sintered Materials

Table 3 shows the average chemical composition of sintered materials (number of tested samples = 5) for a reactive grinding time of 5 h. The chemical composition was determined by inductive coupling plasma spectroscopy (ICP).

The XRD diffraction patterns of the sintered materials at 1300 °C are shown in Figure 2. Table 4 shows the quantitative mineral composition, calculated by the Rietveld method of sintered materials with each type of dross, for different molar relationships. An increase is observed in calcium aluminate content and a decrease in silicate content when the molar ratio CaO/Al_2_O_3_ increases. Put differently, an increase in the system’s calcium content favors the reaction of this element with aluminum, to the detriment of the reaction with silicon.

Using molar ratios 1:2 and 1:3, there is also a transformation in the nature of the aluminates obtained. The disappearance of both monocalcium aluminate (CA) and of most of the formation of tricalcium aluminate (C_3_A), which appears as the most prevalent phase in all sinters, has been observed. This is due to the increased diffusion of the Ca^2+^ within Al_2_O_3_ according to the reaction that summarizes the formation process mechanism:A + C → AC + C → C_12_A_7_ + C → C_3_A.(5)

It is possible to verify how the increase in CaO (C) in the system transforms Al_2_O_3_ (A) in monocalcium aluminate, which is subsequently transformed into C_12_A_7_ and perhaps into other intermediate aluminates, and finally in tricalcium aluminate (C_3_A).

Figure 3 shows the different major mineralogical phases (aluminates) and other minority phases that exist in the sintered materials obtained from each of the studied drosses, for a reactive grinding time of 5 h. The different phases of these sintered materials have been indicated in Table 4, in different colors (orange—material Al 3S, green—Al3 2S and blue—Al3 3S). The results obtained on the mineralogical composition of the sintered materials by XRD are in line with the results of the chemical composition of the sintered materials shown in Table 3.

The mineral phases in the sintered materials are identified through backscattered electrons and chemical microanalysis. In sintered Al3 2S and Al3 3S, the majority phases are calcium aluminates (calcium trialuminate—C_3_A and mayenite—C_12_A_7_), especially the calcium trialuminate in sintered Al3 3S; while in sintered Al3 S, most of the sample studied is made up of calcium aluminate—CA and mayenite—C_12_A_7_, without the appearance of calcium trialuminate—C_3_A, thus confirming the results shown in Table 4, the mineralogical phases obtained by XRD (calculated by the Rietveld method) of the sintered materials are shown.

The transformations produced in the CaO–Al_2_O_3_–SiO_2_ system are shown in Figure 4, where the compositional changes of sintered materials can be observed, due to the increased content of CaO in the mixtures with the drosses.

The sintered materials obtained are within the area of chemical compositions of synthetic drosses indicated by Richardson (1974) [32] as suitable for use in steel manufacturing, especially for its desulfuring effect. At the same time, the sinters obtained, with about 2% MgO content, are of added value, since this compound has a favorable effect on the protection of refractory materials.

Figure 5 shows the Raman spectrum of aluminate Al-3 3S (87% C_3_A and 5.3% C_12_A_7_), obtained from dross Al13 with a molar ratio CaO/Al_2_O_3_ of 1:3. The characteristic peaks of the tricalcium aluminate’s (C_3_A) cubic structure are manifest: a peak centered at 508 cm^−1^ associated with the symmetrical movement of Al–O–Al chemical bonds (ν_1_ [AlO_4_^5−^]). The second peak is centered at 757 cm^−1^ and is assigned to asymmetric narrowing of the AlO_2_ group (ν_3_ [AlO_4_^5−^]). The weak peak that appears at 326 cm^−1^ could indicate the presence of the C_12_A_7_ phase in the analyzed material, since it is characteristic of this phase and it originates in vibrations of the Ca–O system. This result is consistent with the crystalline composition obtained by XRD (see Table 4). A band appears at 358 cm^−1^ associated with the CA–O vibrations in the C_3_A system.

Figure 6a shows a representative TEM image of the obtained Al-1 3S, in which a high degree of crystallinity can be observed. The good crystal quality is further assessed by the SAED (selected area electron diffraction) pattern of the monocrystalline particle, also shown in Figure 6a, where its zone axis is parallel to the [0001] direction of the C_3_A structure. Distances between {200} planes were measured as 7.51 Å. Distances between {030} planes were measured as 5.3 Å. The lattice parameter was extracted directly from Figure 6b, with a value of 2.66 Å, which is in agreement with the typical value for this parameter in C_3_A crystals.

## 4. Conclusions

It is possible to obtain calcium aluminates from the drosses studied using reactive grinding and hot sintering (1300 °C) with calcium carbonate as a precursor. There is a clear relationship between “dross age” (storage time) and the content and nature of the existing aluminates in the sintered materials. The highest content in aluminates is obtained from dross Al-3, the most recent sample, since it has a lower aluminum hydrate content. There is a direct relationship between the contribution of CaO and the aluminate content obtained in the sintered material. The best results are obtained in a CaO-enriched system with a molar ratio Al_2_O_3_/CaO equal to 1:3. In such a system, sintered materials with 90%–92% aluminates are obtained, starting from the most recent drosses Al-3 and Al-1, and about 75% if starting from the oldest dross (Al-2). It is possible to obtain a sintered product with a high tricalcium aluminate content (between 85% and 87%), for use as a synthetic dross in metallurgy. The process of obtaining aluminates is a simple, three-step process: reactive grinding, briquetting and sintering.

## 5. Patents

Method for obtaining calcium aluminates from non-saline aluminum slags. López Gómez, F.A.; Alguacil Prego, F.J.; Ramírez Zablah, M.S. and González Gracia, J.R. PCT/ES2016/070566. WO/2017/017304. 02.02.2017. Available online: https://patentscope.wipo.int/search/en/detail.jsf?docId=WO2017017304 (accessed on 31 March 2018).

## Figures and Tables

**Figure 1 materials-12-01837-f001:**
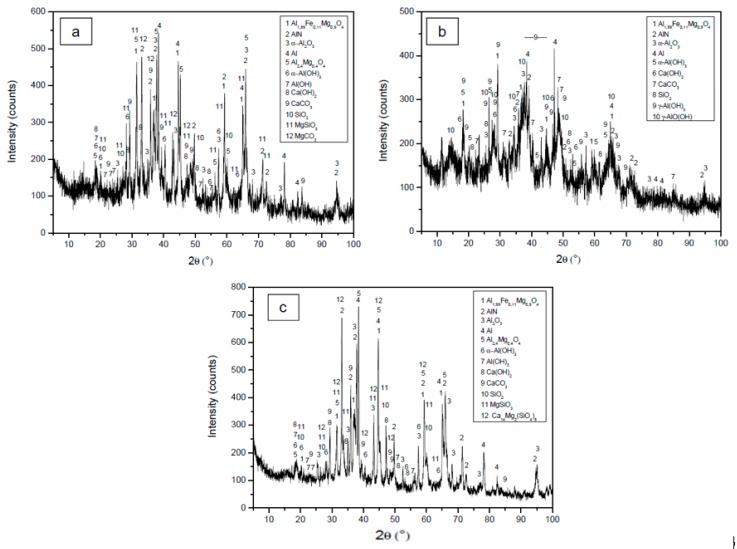
XRD diffraction pattern of the drosses studied; (**a**) Al-1, (**b**) Al-2, and (**c**) Al-3.

**Figure 2 materials-12-01837-f002:**
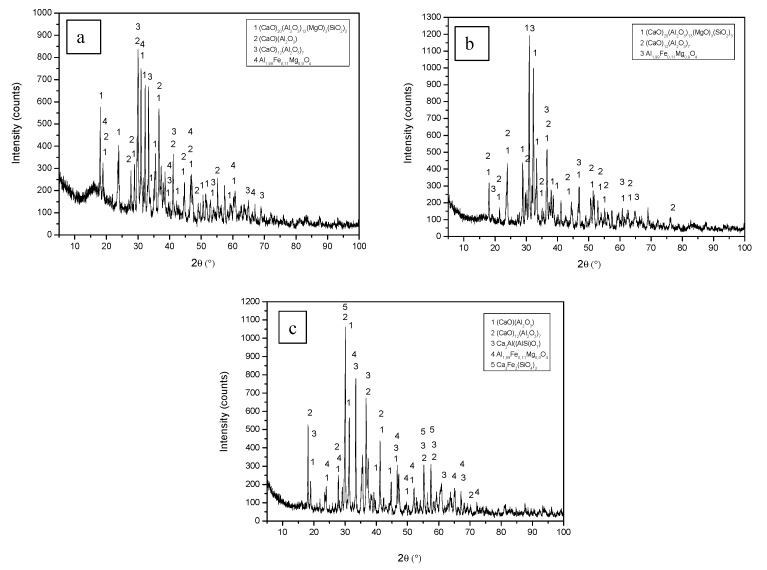
XRD diffraction pattern of the sintered materials at 1300 °C; (**a**) Al_2_O_3_/CaO 1:1 Al1 S; (**b**) Al_2_O_3_/CaO 1:1 Al2 S y; (**c**) Al_2_O_3_/CaO 1:1 Al3 S.

**Figure 3 materials-12-01837-f003:**
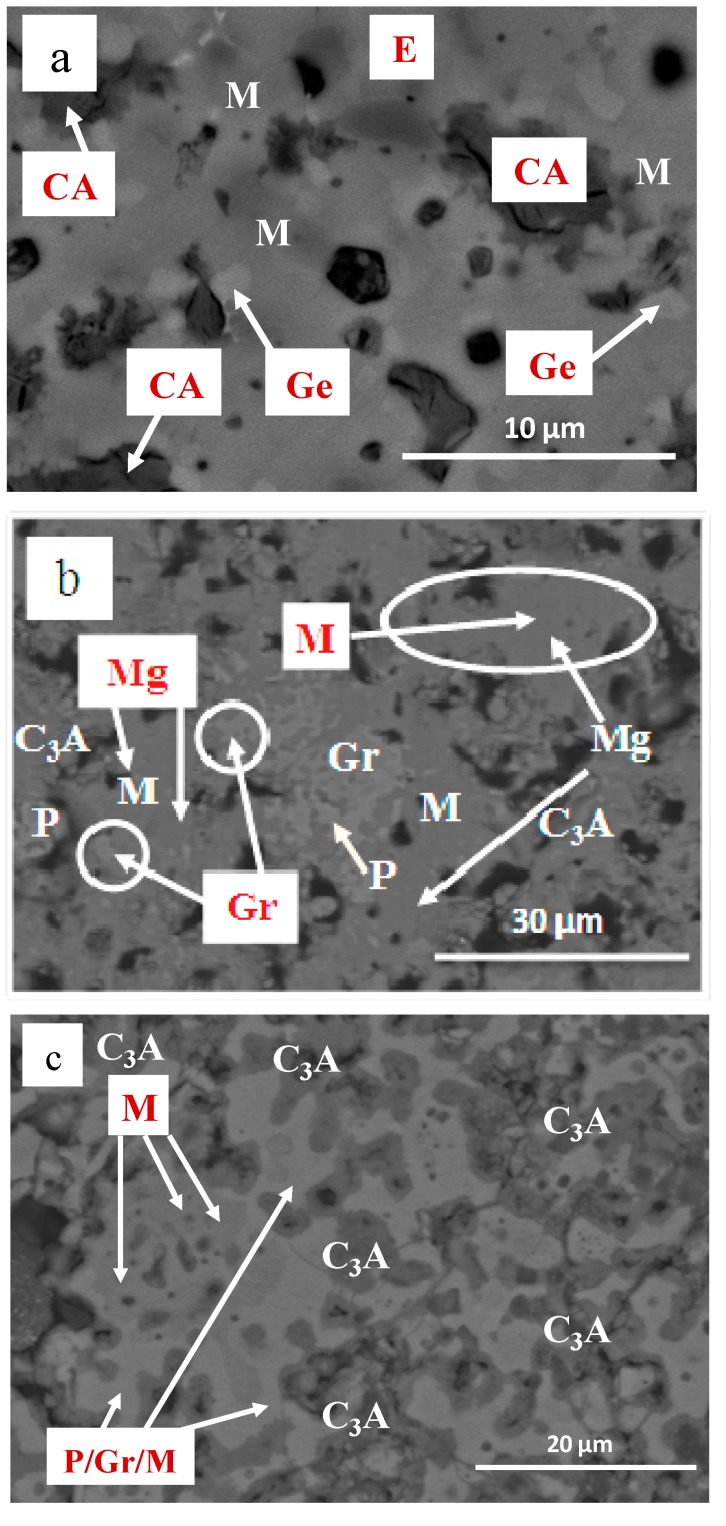
SEM imagen (backscattered electrons) of sinters obtained at 1300 °C; (**a**) Al_2_O_3_/CaO 1:1 Al3 S; (**b**) Al_2_O_3_/CaO 1:2 Al3 2S y (**c**) Al_2_O_3_/CaO 1:3 Al3 3S. (CA = calcium aluminate, M = mayenite or C_12_A_7_, E = spinel, Ge = Al_2_Ca_2_O_7_Si, C_3_A = calcium trialuminate, Gr = Ca_3_Al_2_(SiO_4_)_3_, Mg = MgO y P = Al_1.95_Fe_0.49_Mg_2.65_O_12_Si_2.91_).

**Figure 4 materials-12-01837-f004:**
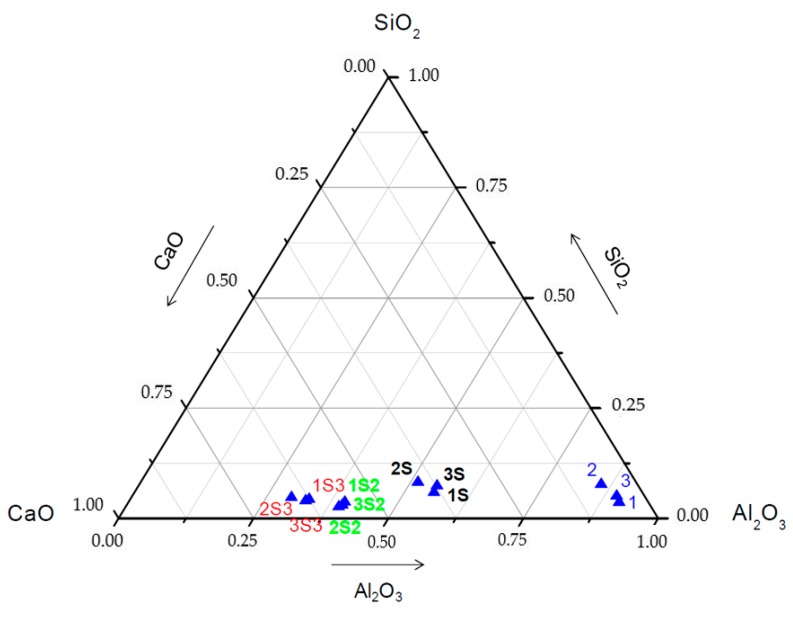
Diagram of phases of the Al_2_O_3_–SiO_2_–CaO system showing the initial drosses (points 1, 2, and 3), the sintered phases with molar ratio CaO/Al_2_O_3_ 1:1 (points 1S, 2S, and 3S), CaO/Al_2_O_3_ 1:2 (points 1S2, 2S2 and 3S2) and sintered phases with molar ratio CaO/Al_2_O_3_ 1:3 (points 1S3, 2S3, and 3S3).

**Figure 5 materials-12-01837-f005:**
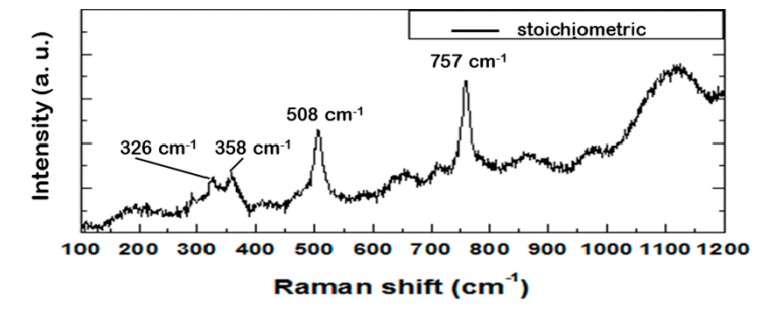
Raman spectra of the obtained Al-1 3S powders excited by a 633 He–Ne laser.

**Figure 6 materials-12-01837-f006:**
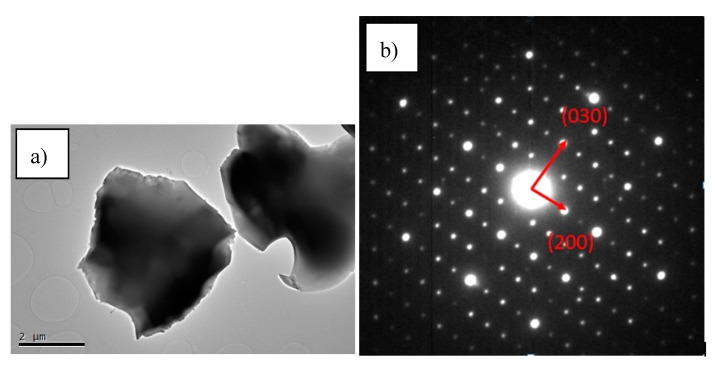
(**a**) TEM image of the Al-1 3S particles; (**b**) shows the SAED of the squared area.

**Table 1 materials-12-01837-t001:** Chemical composition of the studied aluminum drosses (wt %).

Element	Al-1	Al-2	Al-3
Al	40.1	30.9	43.4
Ca	3.2	3.3	3.4
Fe	2.6	3.2	1.3
Mg	1.9	1.2	2.0
Si	1.4	2.5	2.1
Mn	0.19	0.13	0.2
Cu	0.11	0.32	0.08
Zn	0.032	2.018	0.043
Ni	0.020	0.020	0.010
L.O.I.	7.4	17.4	3.3

**Table 2 materials-12-01837-t002:** Mineralogical composition of the aluminum drosses (wt %).

Mineralogical Phase	Al-1	Al-2	Al-3
Spinel, Al_1.99_Fe_0.11_Mg_0.9_O_4_	23.3	13.6	24.2
Aluminum nitride, AlN	13.9	3.1	12.3
Corundum, α-Al_2_O_3_	8.3	6.2	12.0
Metallic aluminum, Al	11.4	3.8	14.4
Spinel, Al_2.4_Mg_0.4_O_4_	23.3	-	15.8
Bayerite, α-Al(OH)_3_	5.9	3.4	2.1
Norstrandite, Al(OH)_3_	1.8	-	0.90
Portlandite, Ca(OH)_2_	1.4	2.2	2.9
Calcite, CaCO_3_	8.7	10.3	6.4
Quartz, SiO_2_	0.80	1.0	0.40
Enstatite, MgSiO_3_	0.80	-	4.5
Magnesite, MgCO_3_	0.60	-	-
Boehmite, γ-AlO(OH)	-	50.4	-
Gibbsite, γ-Al(OH)_3_	-	5.9	-
Bredigite, Ca_14_Mg_2_(SiO_4_)_8_	-	-	3.6

**Table 3 materials-12-01837-t003:** Average chemical composition (number of samples = 5) of the sintered materials (wt %), (A = Al_2_O_3_ and C = CaO).

Element	Al1 S A:C 1:1	Al2 S A:C 1:1	Al3 S A:C 1:1	Al1 2S A:C 1:2	Al2 2S A:C 1:2	Al3 2S A:C 1:2	Al1 3S A:C 1:3	Al2 3S A:C 1:3	Al3 3S A:C 1:3
Al	27.4	26.1	30.0	22.2	20.2	20.7	16.9	15.3	18.4
Fe	1.9	1.6	1.51	1.2	1.75	0.72	0.95	1.13	0.61
Ca	25.6	28.3	27.3	42.1	39.9	39.9	44.2	45.7	46.7
Mg	1.2	0.85	1.1	0.65	0.85	0.89	0.72	0.56	0.80
Si	2.5	3.7	3.5	1.8	1.2	2.4	1.8	2.2	2.1
Mn	0.13	0.09	0.12	0.07	0.11	1.4	0.08	0.05	0.07
Ni	0.03	0.04	0.03	0.02	0.02	0.01	0.02	0.02	0.01
Cu	0.09	0.28	0.07	0.23	0.05	0.03	0.07	0.16	0.11
Zn	0.24	2.25	0.13	1.92	2.2	0.10	0.18	1.5	0.09

**Table 4 materials-12-01837-t004:** Mineralogical composition of the sintered materials with each dross (wt %).

Mineralogical Phase Molar Ratio Al_2_O_3_/CaO	Al1 S 1:1	Al2 S 1:1	Al3 S 1:1	Al1 2S 1:2	Al2 2S 1:2	Al3 2S 1:2	Al1 3S 1:3	Al2 3S 1:3	Al3 3S 1:3
C_3_A*	-	-	-	49.4	49.8	39.2	85.1	71.6	87.0
C_12_A_7_*	19.7	13.3	23.6	32.5	30.6	41.5	5.20	3.75	5.27
CA*	32.8	-	47.8	-	-	-	-	-	-
**Total Aluminates**	**52.5**	**13.3**	**71.5**	**81.9**	**80.5**	**80.7**	**90.2**	**75.4**	**92.2**
Al_1.99_Fe_0.11_Mg_0.90_O_4_	5.5	7.0	10.2	-	-	-	-	-	-
Ca_20_Mg_3_Al_26_Si_3_O_68_	41.5	79.6	-	-	-	-	-	-	-
Ca_3_Fe_2_(SiO_4_)_3_	-	-	5.3	-	-	-	-	-	-
Al_2_Ca_2_O_7_Si	-	-	14.5	-	-	-	-	-	-
Al_1.95_Fe_0.49_Mg_2.65_O_12_Si_2.91_	-	-	-	2.58	10.57	1.66	2.29	9.68	1.78
Ca_3_Al_2_(SiO_4_)_3_	-	-	-	13.44	-	15.96	2.13	1.85	1.40
Ca_6_(SiO_4_)(Si_3_O_10_)	-	-	-	-	7.03	-	-	9.01	-
Al_0.2_Fe_1.8_MgO_4_	-	-	-	-	1.96	-	-	0.97	-
**Silicates and Other Phases**	**47.0**	**86.5**	**29.6**	**16.02**	**19.86**	**17.62**	**4.4**	**21.6**	**3.1**
SiO_2_	-	-	-	-	-	-	-	0.25	-
CaO	-	-	-	0.63	-	-	3.28	2.85	2.25
MgO	-	-	-	1.50	-	1.68	2.05	-	2.32

(* C = CaO and A = Al_2_O_3_).

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
