# Peer review of "Synthesis of Calcium Aluminates from Non-Saline Aluminum Dross"

_materials, 2019, doi:10.3390/ma12111837_

Round 1
Reviewer 1 Report
It is an interesting and valuable piece of work. Please remove the existing in th text Spanish words, e.g. the line no. 201 (puntos; y), 211 (DRX), Fig. 3 (u.a. ; stequiometric). Please correct numbers (subscripts) in some chemical formulas. Please correct the line no. 50.
Author Response
Thank you for sending us the Referees’ comments on our paper; they were very useful and have helped us improve the quality of our manuscript. We would like to thank you and the Referees for the efforts made in examining our work.
All suggestions from the reviewer have been included in the revised manuscript and indicated in yellow in the text
Reviewer 2 Report
The authors present a well-written and structured thorough documentation of the dross produced in the metal-aluminium fusion plant. The scientific methods are sound, the results are properly discussed including the statistics where possible. For that reason the paper should be accepted after addressing a few minor remarks:
1. in some sections the authors actually are able to present average results (e.g. average chemical composition). It would be helpful to provide each time the number of tested samples.
2. Should be changed page 7, L189: mayenita -> mayenite.
3. I would change: calcium trialuminate – C3Al -> C3A which is more commonly used, view full text.
4. It would be appreciated if the authors tried to link more the results of the XRD etc with consequences for the chemical composition and microstructural characterization of sintered materials.
Author Response
Thank you for sending us the Referees’ comments on our paper; they were very useful and have helped us improve the quality of our manuscript. We would like to thank you and the Referees for the efforts made in examining our work.
The four suggestions of the reviewer have been included in the revised manuscript and have been indicated in yellow in the text
Reviewer 3 Report
The manuscript can be improved considering the following points:
In table 1, which is the chemical composition of the dross measured by ICP method, the components are all reported as oxides, while the ICP result is elemental analysis. Moreover, in Al dross there is always some metallic Al. Moreover, Zinc can be in the form of metal in Al matrix. If the authors present the elements in the dross instead of oxide form, it is better and it does not affect the results and discussions.
In Table 2, the insignificant compounds with not detectable peaks in XRD (the last 9 compounds) must be removed. Presenting the XRD spectrum of the dross is also supportive and recommended.
The presented data in Tables 3 and 4 are confusing. The XRD results in Table 4 and SEM analysis show different compunds that are listed in Table 4. However, the data in Table 3 show that in the same materials the components are all oxides! It is necessary in section 3.2 to explain the way the data in Tables 3 and 4 are obtained, and the proper way applied.
The addition of XRD results for sintered samples in the manuscript is needed to support the presented data in Table 4.
In line 109, the type of mill must be mentioned. TEMA seems to be a brand name/manufacturer!
In line 137, the purity of CaCO3 must be given.
The text can be improved and some corrections can be done, such as:
- line 74: Ca2 must be CA2
- line 76: Ca6 must be CA6
- line 183: C3Al must be C3A
- line 188: Imagen must be image
- line 211: DRX must be XRD
-
Author Response
In table 1, which is the chemical composition of the dross measured by ICP method, the components are all reported as oxides, while the ICP result is elemental analysis. Moreover, in Al dross there is always some metallic Al. Moreover, Zinc can be in the form of metal in Al matrix. If the authors present the elements in the dross instead of oxide form, it is better and it does not affect the results and discussions.
Table 1 has been modified, indicating the elemental composition obtained by ICP
In Table 2, the insignificant compounds with not detectable peaks in XRD (the last 9 compounds) must be removed. Presenting the XRD spectrum of the dross is also supportive and recommended.
Table 2 has been modified following the suggestions of the reviewer. A new figure has been included (Fig.1) that includes the diffractograms of the studied dross
The presented data in Tables 3 and 4 are confusing. The XRD results in Table 4 and SEM analysis show different compunds that are listed in Table 4. However, the data in Table 3 show that in the same materials the components are all oxides! It is necessary in section 3.2 to explain the way the data in Tables 3 and 4 are obtained, and the proper way applied.
Table 4 shows the results obtained in the determination of mineralogical composition by Rietveld analysis. Lines 168-169 have been clarified in the text
A new Figure (Fig.2) has been included with the diffractograms of the sintered samples. In this way, Table 4 is supported by the information in Fig.2
The addition of XRD results for sintered samples in the manuscript is needed to support the presented data in Table 4.
The figure has been included (Fig.2)
In line 109, the type of mill must be mentioned. TEMA seems to be a brand name/manufacturer!
It has been corrected and marked in red
In line 137, the purity of CaCO3 must be given.
It has been corrected and marked in red
The text can be improved and some corrections can be done, such as:
- line 74: Ca2 must be CA2
It has been corrected and marked in red
- line 76: Ca6 must be CA6
It has been corrected and marked in red
- line 183: C3Al must be C3A
It has been corrected and marked in red
- line 188: Imagen must be image
It has been corrected and marked in red
- line 211: DRX must be XRD
It has been corrected and marked in red

Round 2
Reviewer 3 Report
Regarding the mentioned point in line 174 and table 3:
In principle, XRD analysis provides information about the compounds in the samples, the results that are given in Table 4. The results in Table 3 cannot be obtained directly by any characterization method. If the samples contain oxides that are given in Table 3, then they cannot be in the form of compounds in Table 4. The question is that how the information in Table 3 have been obtained? ICP analysis cannot provide the information in Table 3. The authors must clearly explain in the manuscript how the information in Tables 3 have been obtained, or remove Table 3.
Author Response
In principle, XRD analysis provides information about the compounds in the samples, the results that are given in Table 4. The results in Table 3 cannot be obtained directly by any characterization method. If the samples contain oxides that are given in Table 3, then they cannot be in the form of compounds in Table 4. The question is that how the information in Table 3 have been obtained? ICP analysis cannot provide the information in Table 3. The authors must clearly explain in the manuscript how the information in Tables 3 have been obtained, or remove Table 3.
Table 3 showing the elemental chemical composition of the sintered materials obtained by ICP has been corrected. Fig. 2 shows the diffractograms of the sintered materials. From these results and by means of the Rietlved analysis, the results shown in Table 4 are obtained. In this way, both results are coherent. It has been clarified in the text, in lines 196-204.